# Dose- and Time-Dependent Effects of Radiofrequency Electromagnetic Field on Adipose Tissue: Implications of Thermoregulation and Mitochondrial Signaling

**DOI:** 10.3390/ijms241310628

**Published:** 2023-06-25

**Authors:** Jennifer Maalouf, Amandine Pelletier, Aurélie Corona, Jérôme Gay-Quéheillard, Véronique Bach, René de Seze, Brahim Selmaoui

**Affiliations:** 1PériTox-Périnatalité et Risques Toxiques-UMR_I 01, Centre Universitaire de Recherche en Santé, CURS-UPJV, University of Picardy Jules Verne, CEDEX 1, 80054 Amiens, France; amandine.pelletier@u-picardie.fr (A.P.); aurelie.corona@u-picardie.fr (A.C.); jerome.gay@u-picardie.fr (J.G.-Q.); veronique.bach@u-picardie.fr (V.B.); rene.de.seze@wanadoo.fr (R.d.S.); brahim.selmaoui@ineris.fr (B.S.); 2Department of Experimental Toxicology and Modeling (TEAM), Institut National de l’Environnement Industriel et des Risques (INERIS), 60550 Verneuil-en-Halatte, France

**Keywords:** radiofrequency, thermoregulation, adipose tissue, mitochondria

## Abstract

Recent studies have shed light on the effects of low-intensity radiofrequency (RF) fields on thermoregulation and adipose tissue metabolism. The present study aims to further explore these effects by analyzing the expression of thermoregulatory genes and investigating the involvement of mitochondria in adipose tissue metabolism. Male mice (n = 36 C57BL/6J) were assigned to either exposed or control groups. The exposed groups were subjected to RF fields at 900 MHz, with specific absorption rates (SAR) of 0.1 W/kg or 0.4 W/kg, either for three or seven consecutive days. The findings indicate that RF exposure leads to changes in adipose tissue markers, with some effects being dose-dependent and time-dependent. In brown adipose tissue (BAT), after 3 days of RF exposure, thermogenesis is reduced, mitochondrial activity in BAT decreases, and an increase in gene expression, responsible for balancing the regulatory and damaging effects of reactive oxygen species (ROS), was observed. This effect was partially compensated after 7 days of exposure. In white adipose tissue (WAT), RF exposure results in reduced fatty acid oxidation, impaired energy production, and hindered adipocyte differentiation. Notably, no effects of RF on mitochondrial biogenesis in WAT were observed. These findings contribute to understanding the effects of RF exposure on adipose tissue metabolism and thermoregulation, highlighting dose-dependent and time-dependent responses.

## 1. Introduction

Over the past few decades, there has been an increasing concern about the potential health effects of electromagnetic fields, particularly those stemming from the widespread use of wireless devices, such as cell phones and WiFi routers. While the majority of attention has focused on the potential health effects from exposure to high levels of radiofrequency fields, emerging evidence suggests that low-intensity radiofrequency fields may also exert non-thermal effects on biological systems. One area garnering interest is the impact of radiofrequency fields on thermoregulation.

One important aspect to consider when examining the effects of low-intensity RF fields on biological systems is the dose-response relationship. This concept can provide critical information on the potential health risks associated with low-intensity RF exposure, and can inform the development of safety guidelines and regulations for the use of wireless devices. It has been observed that the thermal effects of RF fields are typically observed at exposure levels of 4 W/kg and above. An increase in body temperature greater than 1 °C can occur when the entire body is exposed to RF fields with SAR values exceeding 4 W/kg [1]. Furthermore, studies have investigated the interactions of biological systems with electromagnetic radiation (EMR), specifically focusing on the effects of microwaves on the skin and reproductive system, and analyzing the impact of microwave radiation on the brain [2].

At non-thermal levels, at a SAR below 4 W/kg, a series of studies conducted by Arendash’s team investigated the impact of 918 MHz RF exposure at an estimated SAR of 0.25 W/kg on mice. They found that significant increases in body temperature were observed only after chronic and repeated exposure, suggesting a physiological activation related to thermogenesis, a heat-producing response in animals exposed to cold [3,4]. In other studies examining the effects of RF exposure on temperature regulation and thermal preference in animals, several key findings emerged. One set of studies on juvenile male Wistar rats exposed to 900 MHz RF fields at 1 V/m for 23 h a day over 5 weeks found that exposed rats exhibited lower tail temperatures and altered thermal preferences compared to controls, which resembled the responses of animals exposed to cold environments [5,6]. In a study exposing mice to 900 MHz RF at a SAR of 0.16 W/kg twice daily for one hour in both the morning and afternoon for seven consecutive days, a significant increase in body temperature was observed, starting on the fourth day of exposure, during the exposure, and within one hour of the end of RF exposure. These findings highlight the potential impact of RF exposure on temperature regulation and thermogenesis in animals [7], and suggest an activation of thermogenesis in the exposed mice. Therefore, an effect of RF on mouse thermogenesis has been hypothesized, and the biological and physiological mechanisms involved in the measured temperature variations need to be investigated.

Thermoregulation is a vital physiological process that enables mammals to maintain a constant body temperature despite changes in the environment. BAT is a specialized type of adipose tissue that plays a critical role in thermoregulation by generating heat through non-shivering thermogenesis (NST) [8]. The non-shivering thermogenesis process is initiated by the uncoupling of oxidative phosphorylation in mitochondria, which is regulated by several key enzymes and proteins. One such protein is type 2 iodothyronine deiodinase (DIO2), which has been identified as playing a critical role in BAT thermogenesis. DIO2 is an enzyme that converts the inactive thyroid hormone T4 into its active form, T3 [9]. In BAT, DIO2 has been shown to be critical for the induction of thermogenesis, as it regulates the availability of T3 for binding to nuclear receptors and allows for the subsequent activation of thermogenic genes, such as uncoupling protein 1 (UCP1) [10]. Mitochondria are the primary site of energy production in the cell, and they play a crucial role in thermogenesis in BAT. During NST, the activation of UCP1 in the inner mitochondrial membrane leads to the uncoupling of oxidative phosphorylation, resulting in the production of heat instead of ATP [11]. UCP1 is a unique protein that is expressed in BAT mitochondria and functions as a proton transporter that dissipates the proton gradient generated by the electron transport chain [12]. UCP1 is essential for BAT thermogenesis and is a key mediator of the thermogenic response to cold exposure and other stimuli [11]. On the other hand, citrate synthase (CS) is a key enzyme in the tricarboxylic acid (TCA) cycle, which is responsible for the production of ATP through the process of oxidative phosphorylation. In BAT, CS plays an important role in regulating the production of heat through the TCA cycle. Specifically, CS activity has been shown to be positively correlated with the activity of UCP1 [13]. ACO1, or aconitase, is an enzyme involved in the regulation of cellular metabolism and iron homeostasis. Aconitase has two forms: cytosolic aconitase (cAco) and iron regulatory protein 1 (IRP1). The cytosolic isoform functions as an enzyme in its reduced form and as IRP1 in its oxidized form, regulating iron homeostasis. Aconitase enzymes, including ACO1, play a role in balancing the regulatory and damaging effects of ROS, which are byproducts of cellular metabolism. By contributing to the regulation of cell metabolism and iron homeostasis, aconitase enzymes help maintain the balance between ROS production and cellular protection [14]. In addition, the expression of carnitine palmitoyltransferase-1 CPT1α plays a key role in fatty acid metabolism by facilitating the transport of long-chain fatty acids into mitochondria for oxidation [15]. In BAT, CPT1α has been shown to be critical for the induction of thermogenesis, as it regulates the availability of fatty acids for oxidation and subsequent heat generation [16].

While BAT is primarily responsible for thermoregulation in mammals, recent studies have also highlighted the potential role of WAT in this process. WAT is a major tissue for energy storage in the body and is traditionally viewed as a passive tissue that stores excess calories in the form of triglycerides. However, studies have shown that WAT contributes to the regulation of body temperature through the production of heat in response to cold exposure, a process known as “browning” of white adipocytes [17]. When WAT undergoes the browning process, it becomes more metabolically active and contributes to thermoregulation by generating heat. The browning process undergoes various changes, including greater energy expenditure, fat multilocularization, an increase in mitochondrial numbers, and an activation of mitochondrial biogenesis and remodeling, resulting in increased mitochondrial density, size, and oxidative capacity. In browning WAT, CS expression is upregulated, indicating an increase in TCA cycle activity and mitochondrial metabolism [18]. The increased CS expression is also accompanied by upregulation of other mitochondrial enzymes, such as acetyl-CoA synthase and NADH:ubiquinone oxidoreductase core subunit S8 (NDUFS8) (a subunit of mitochondrial complex I) [19], which enhance fatty acid oxidation and electron transport chain activity, respectively. In addition, the expression of CPT1α is also increased, facilitating fatty acid uptake and β-oxidation in the mitochondria. These changes collectively enhance mitochondrial function and contribute to thermogenesis in browning WAT [16].

Through the browning process, WAT undergoes various changes, including an increase in mitochondrial numbers and activity, greater energy expenditure, fat multilocularization, and expression of specific thermogenic genes [20]. These genes include peroxisome proliferator-activated receptor-a (PPARα) and PR domain-containing 16 (PRDM16), calcium-binding protein S100b, and UCP1. UCP1 is primarily expressed in BAT, but recent studies have shown that it can also be expressed in WAT in response to various stimuli, including cold exposure and PPARα activation. PPARα is a transcription factor that plays a critical role in lipid metabolism and energy homeostasis. Activation of PPARα in WAT leads to increased expression of UCP1 and increased thermogenesis, which can contribute to energy expenditure and weight loss. In addition to its role in thermogenesis, PPARα also regulates fatty acid metabolism in WAT, promoting fatty acid oxidation and decreasing lipid storage. PRDM16 promotes browning by activating the expression of brown fat-specific genes, leading to increased mitochondrial biogenesis, increased oxidative metabolism, and increased energy expenditure. In addition, PRDM16 can interact with other transcriptional regulators, such as PPARα and PPARδ, to induce a brown fat-like phenotype in WAT. PRDM16 controls S100B, a calcium-binding protein that has been shown to play a role in adipose tissue metabolism and thermogenesis. S100B levels were increased in the adipose tissue of mice subjected to cold exposure, which is known to induce browning of WAT [21,22,23,24,25]. Finally, there is emerging evidence to suggest that the SLC6A8 pathway may serve as an alternative thermogenic pathway in WAT thermoregulation, in addition to the well-established UCP1 pathway. Recent studies have demonstrated that the genetic deletion of SLC6A8 in mice leads to impaired thermogenic capacity in WAT, suggesting that SLC6A8 plays an important role in WAT thermoregulation [25].

Despite some research on the potential dose-dependent effects of RF fields on biological systems, the underlying mechanisms are still not well understood. Therefore, further research is crucial to investigate the dose-response effects of low-intensity RF fields on thermoregulation. Specifically, it is important to understand the mechanisms involved in the observed effects on thermoregulation and to use markers to evaluate if a dose-response effect exists that is similar to the thermal effect observed at exposure levels exceeding 4 W/kg. This study aims to examine the dose-response effects of low-intensity RF fields on adipose tissue by analyzing the expression of thermoregulatory genes and the involvement of mitochondria in these effects. To ensure relevance to real-world exposure scenarios, we selected an RF wavelength of 900 MHz, commonly encountered in everyday wireless communication devices. Furthermore, our choice of SAR values below 0.4 W/kg, which is well below the thermal threshold of 4 W/kg, allowed us to investigate the non-thermal effects of RF radiation. By exploring different intensities within the non-thermal range, such as the SAR of 0.1 W/kg, we aimed to uncover potential variations in the response of adipose tissue to RF radiation. The exposure duration of 1 h twice a day was based on previous studies that reported effects of low-intensity RF radiation on thermoregulation.

## 2. Results

### 2.1. Dose-Response Effect of 3 Days of RF Radiation on Adipose Tissue

#### 2.1.1. Gene Expression in BAT

After three days of exposure to RF fields, the expression of DIO2 mRNA in BAT showed a significant downregulation in the RF-exposed group at 0.1 W/kg compared to the sham group, whereas no significant difference was observed in the RF-exposed group at 0.4 W/kg. Similar findings were observed for CS mRNA. In terms of UCP1 mRNA expression, both the RF-exposed groups at 0.1 W/kg and 0.4 W/kg exhibited a significant downregulation compared to the sham group. On the other hand, the expression of ACO1 mRNA was significantly upregulated in the RF-exposed group at 0.1 W/kg compared to the sham group, while no significant difference was observed in the RF-exposed group at 0.4 W/kg. Interestingly, there was no significant effect observed on the expression of CPT1α in BAT in either of the RF-exposed groups compared to the sham group (Figure 1).

#### 2.1.2. Parameters Analyzed in WAT

A.Gene Expression

After three days of RF exposure, the S100b, PPARα, and CS mRNA expression was found to be significantly downregulated in WAT of the RF-exposed group at 0.1 W/kg compared to the sham group, but there was no significant difference in the RF-exposed group at 0.4 W/kg. In contrast, the PRDM16 mRNA expression was significantly upregulated in WAT of the RF-exposed group at 0.1 W/kg compared to the sham group, while no significant difference was observed at 0.4 W/kg. Furthermore, the CPT1α mRNA expression was significantly upregulated in WAT of the RF-exposed group at 0.4 W/kg, but there was no significant difference at 0.1 W/kg. However, there was no significant effect on the expression of ACO1, NDUFS8, and Slc6a8 in WAT in the RF-exposed groups compared to the sham group (Figure 2).

B.Mitochondrial Quantification

The impact of three days of RF exposure on mitochondrial DNA (mtDNA) copy number was evaluated at 0.1 W/Kg and 0.4 W/Kg. Our results indicate that there was no significant effect on the mtDNA copy number at either 0.1 W/Kg or 0.4 W/Kg, with respect to the control group, after RF exposure (Figure 3).

### 2.2. Dose-Response Effect of 7 Days of RF Radiation on Adipose Tissue

#### 2.2.1. Gene Expression in BAT

After seven days of RF exposure, the expression of CS mRNA in BAT was found to be significantly upregulated in the RF-exposed group at 0.4 W/kg compared to the sham group. However, there was no significant difference observed in the RF-exposed group at 0.1 W/kg. Conversely, the DIO2 mRNA expression was found to be significantly downregulated in the RF-exposed group at 0.1 W/kg compared to the sham group, while no significant difference was observed in the RF-exposed group at 0.4 W/kg. Interestingly, there was no significant effect observed on the expression of CPT1α, UCP1, and ACO1 in BAT in either of the RF-exposed groups compared to the sham group (Figure 4).

#### 2.2.2. Parameters Analyzed in WAT

A.Gene Expression

After seven days of exposure, the mRNA expression of PPARα, ACO1, and S100b was found to be significantly downregulated in the RF-exposed group at both 0.1 W/kg and 0.4 W/kg when compared to the sham group. In contrast, the mRNA expression of PRDM16 and Cs were significantly downregulated in the RF-exposed group at 0.1 W/kg compared to the sham group, but there was no significant difference with the RF-exposed group at 0.4 W/kg. Furthermore, the expression of Slc6a8 mRNA in WAT was found to be significantly upregulated in the 0.4 W/kg RF-exposed group, but there was no significant difference in the RF-exposed group at 0.1 W/kg when compared to the sham group. However, there was no significant effect observed on the expression of CPT1α and NDUFS8 in WAT in either of the RF-exposed groups compared to the sham group (Figure 5).

B.Mitochondrial Expression

The effect of 7 days of RF exposure on mitochondrial DNA (mtDNA) copy number was investigated at two different exposure levels, 0.1 W/Kg and 0.4 W/Kg. Our findings revealed that there were no statistically significant differences in the mtDNA copy number between the RF-exposed groups and the control group at either (Figure 6).

## 3. Discussion

Studies have shown that exposure to low-intensity radiofrequency fields can result in altered thermal preferences and changes in body temperature, potentially impacting thermogenesis in animals. Adipose tissue plays a significant role in thermogenesis, particularly BAT, which specializes in energy expenditure and heat production. The presence of beige adipocytes within WAT depots can also be induced to exhibit thermogenic capacity. Consequently, our research investigated the impact of RF exposure on the mRNA expression of various adipose tissue markers in mice.

In BAT, one of the significant findings in the study was the downregulation of DIO2 mRNA expression in the RF-exposed group at 0.1 W/kg after 3 days of exposure, with a similar effect seen after 7 days of exposure. DIO2 encodes an enzyme that converts the thyroid hormone, T4, to the biologically active form, T3. T3 is an important regulator of metabolism, and its production in BAT is necessary for thermogenesis [10]. Thus, the downregulation of DIO2 in BAT can impair the conversion of T4 to T3, which can lead to a decrease in thermogenesis. T3 has been shown to stimulate the expression of UCP1, which uncouples mitochondrial respiration from ATP synthesis, resulting in the generation of heat [26]. UCP1 mRNA expression was downregulated in the RF-exposed group at 0.1 W/kg compared to the sham group. This finding aligns with the understanding that T3 stimulates the expression of UCP1, as mentioned earlier. By uncoupling mitochondrial respiration from ATP synthesis, UCP1 promotes the generation of heat. Therefore, the downregulation of DIO2 can have a cascading effect on UCP1 expression. Kazak et al. suggest that reduced electron transport chain (ETC) activity may be commonly associated with decreased UCP1 levels more generally [27]. In addition to the downregulation of DIO2 and UCP1 mRNA expression observed in our study, we also found a downregulation of CS mRNA expression in the RF-exposed group at 0.1 W/kg after 3 days of exposure. CS is an enzyme involved in the TCA cycle and serves as a marker of mitochondrial activity [28]. The downregulation of CS mRNA expression suggests a potential decrease in mitochondrial function in BAT following RF exposure. Furthermore, our study revealed an upregulation of ACO1 mRNA expression in the RF-exposed group at 0.1 W/kg after 3 days of exposure, suggesting a potential adaptive response to the RF exposure. This response may involve mechanisms aimed at maintaining cellular metabolism and redox balance in the face of altered thermogenic capacity in BAT. The upregulation of ACO1 mRNA expression may indicate a compensatory response to the downregulation of other components related to thermogenesis. Aconitase enzymes, including ACO1, play a role in balancing the regulatory and damaging effects of ROS, which are byproducts of cellular metabolism. By contributing to the regulation of cell metabolism and iron homeostasis, aconitase enzymes help maintain the balance between ROS production and cellular protection [14]. Recent studies have consistently shown that exposure to electromagnetic fields (EMF) induces oxidative stress in different tissues. Additionally, EMF exposure has been associated with significant alterations in blood antioxidant markers [29].

The transient nature of the effect observed after 3 days of exposure at 0.1 W/kg, but not after 7 days, for UCP1 and ACO1 suggests that the cellular response to RF field may compensate after 7 days of exposure. However, the downregulation of DIO2 persisted even after 7 days of exposure in our study. Despite this downregulation, compensatory mechanisms were observed to maintain normal UCP1 expression. Previous studies have shown that mice lacking DIO2 in BAT can exhibit compensatory upregulation of another enzyme called DIO1. DIO1 is an alternate enzyme that can convert T4 to T3. This compensatory upregulation of DIO1 allows for the maintenance of normal levels of T3 production, which is crucial for thermogenesis and the expression of UCP1 [30].

Interestingly, we observed that UCP1 mRNA expression was downregulated in the RF-exposed group compared to the sham group, after 3 days of RF exposure at a SAR of 0.4 W/kg. However, there was no significant effect observed on the expression of other genes in BAT, indicating a potential dose-dependent response, where higher doses may have different, or no, effects on gene expression in BAT. Additionally, after 7 days of exposure at a SAR of 0.4 W/kg, we found an upregulation of CS mRNA expression in the RF-exposed group compared to the sham group. This upregulation indicates a potential adaptation of BAT to the RF exposure, resulting in enhanced mitochondrial activity, which may be related to increased thermogenic capacity.

The findings from our study revealed a downregulation of PPARα mRNA gene expression in WAT after 3 days of RF exposure at 0.1 W/kg. These results are intriguing, considering that PPARα activation has been associated with the induction of brown adipocyte-selective genes and the promotion of browning in WAT. Studies have shown that downregulation of PPARα in WAT may lead to decreased fatty acid oxidation and browning of adipocytes, which could contribute to decreased thermogenesis and energy expenditure [21]. Additionally, our study demonstrated a downregulation of the CS mRNA gene expression in WAT after 3 days of RF exposure at 0.1 W/kg. CS is a key enzyme in the citric acid cycle, responsible for generating ATP, the energy currency of cells. A decrease in citrate synthase activity may result in reduced energy production in white adipose tissue, hinder the beiging process and impairing the tissue’s ability to contribute to thermogenesis. At a SAR of 0.1 W/kg, we observed a downregulation of S100B expression in the RF-exposed group compared to the sham group. This downregulation of S100B expression in the RF-exposed group, at an SAR of 0.1 W/kg, suggests that RF exposure may potentially interfere with adipose tissue metabolism and thermogenesis. Furthermore, our study found a significant upregulation of PRDM16 mRNA expression in the RF-exposed group at 0.1 W/kg after 3 days of exposure. PRDM16 is required for the browning of WAT, and its downregulation has been shown to impair the ability of WAT to undergo browning and generate heat [31]. The upregulation of PRDM16 could be a compensatory response to other changes, or may indicate a potential for increased browning, despite other negative metabolic changes.

After 7 days of RF-exposure at 0.1 W/kg, S100B, PPARα, and CS mRNA expression were still downregulated. Surprisingly, PRDM16 mRNA expression was downregulated in the RF-exposed group compared to the sham group. PRDM16 is required for the browning of WAT, and its downregulation has been shown to impair the ability of WAT to undergo browning and generate heat [31]. Additionally, ACO1 mRNA expression was downregulated in the RF-exposed group compared to the sham group. The knockdown of ACO1 led to reduced cytosolic aconitase activity and diminished intracellular NADPH levels. Consequently, this may cause hindered adipocyte differentiation, smaller lipid droplets in adipocytes, decreased expression of adipogenic, lipogenic, and lipolytic genes, and decreased ATP production [32].

The exposure to RF radiation at an SAR of 0.4 W/kg for three days resulted in an upregulation of CPT1α mRNA expression in WAT. This upregulation suggests a potential enhancement of fatty acid oxidation, leading to increased energy expenditure or other metabolic effects in WAT. Conversely, similar to the effects observed at an SAR of 0.1 W/kg, the expressions of S100B, PPARα, and ACO1 were downregulated. Furthermore, the expression of SLC6A8 was also upregulated. Recent studies have shown that the genetic deletion of SLC6A8 in mice leads to impaired thermogenic capacity in WAT, indicating that SLC6A8 plays an important role in WAT thermoregulation [25]. However, no significant effect on CS was observed. Therefore, it appears that the effects on CS, SLC6A8, and CPT1α may be dose-dependent, varying with different levels of RF exposure.

The observed differences in gene expression and response to RF radiation between BAT and WAT in our study can be attributed to their distinct cellular compositions and metabolic functions. BAT is characterized by a higher abundance of brown adipocytes and higher mitochondrial density and metabolic activity, making it more susceptible to environmental stimuli, such as RF radiation. In contrast, WAT contains predominantly white adipocytes and may exhibit different sensitivities or adaptive responses to RF radiation.

Finally, our study examined the effect of RF exposure on mtDNA copy number in WAT, which is an important indicator of the browning process and essential for thermogenesis. The results showed that RF exposure at 0.1 W/Kg and 0.4 W/Kg did not have a significant effect on mtDNA copy number in WAT after three and seven days of exposure. This suggests that RF exposure did not induce the browning of WAT through an increase in mtDNA copy number.

In conclusion, the findings of this study shed light on the potential effects of RF radiation exposure on adipose tissue. In BAT, the downregulation of CS, UCP1, and DIO2 mRNA expressions in BAT after 3 days of RF exposure at a SAR of 0.1 W/Kg indicates a reduced thermogenesis in BAT. However, compensatory mechanisms, represented by the upregulation of ACO1, were observed, suggesting an adaptation of BAT to RF exposure. In WAT, alterations were observed in the expression of several markers involved in thermogenesis and browning. Decreased expression of PPARα, CS, S100B, and PRDM16 mRNA was observed, indicating potential disruptions in metabolism, browning, and thermogenesis. However, upregulation of CPT1α mRNA expression, observed at a SAR of 0.4 W/kg, suggests a potential increase in fatty acid oxidation.

Overall, these findings highlight the complex effects of RF radiation exposure on adipose tissue, with the potential to either decrease or increase thermogenesis and browning, depending on the SAR level. Further research is needed to better understand the mechanisms underlying these effects and the threshold of this change of effects, with further investigation on the protein level.

## 4. Materials and Methods

### 4.1. Ethics Agreement

The use of animals in this study has been approved by the Regional Directorate for Health, Animal Protection, and the Environment in Amiens, France, which is a nationally accredited organization. The French Ministry of Research has also granted permission for the study, under permit number 21830. The animals have been treated in compliance with both the European guidelines (2010/63/EU) and the French governmental decree 2013–118, which outline the appropriate care and use of laboratory animals.

### 4.2. Animal’s Housing

Three-month-old C57BL/6J mice were purchased from Janvier Lab (Le Genest Saint Isle, France). On arrival to the facility, the mice were housed for five days for acclimation. Animals were housed one per cage (polycarbonate cage1264C, Eurostandard Type II—267 × 207 × 140 mm), with cages having stainless steel lids. The animals were examined daily to ensure their good health. The animals had access to food and filtered tap water ad libitum. All the environmental parameters were controlled: 12/12 h dark/light cycle, lights on at 7:00 a.m. and off at 7:00 p.m., ambient temperature of 22 ± 2 °C, relative air humidity between 30 and 70%.

### 4.3. Exposure System and Dosimetry

Exposures were made in a reverberation chamber setup built by IT’IS (IT’IS Foundation, Zurich, Switzerland) (Capstick et al., 2017). A SMIQ02B generator (Rohde and Schwarz, Munich, Germany) was used to generate radio frequency signals. The generator output signal was amplified with a 700 MHz to 2.2 GHz TE UMS 015 AA amplifier (Zurich, Switzerland), measured and recorded using a data acquisition system. Each chamber contained two stirrers (vertical and horizontal) and three internal 700 MHz to 2.2 GHz wideband log-periodic antennas from SPEAG (Zurich, Switzerland) to excite the reverberant cavity. The antenna was aimed at one of the stirrers to optimize the homogeneity of the RF field inside the chamber. The HF field homogeneity in the horizontal plane, at a distance of 100 mm, in the reverberation chamber was measured by IT’IS (Zurich, Switzerland). The electric field in each chamber was monitored with two ER3DV5 electric field probes (SPEAG, Zurich, Switzerland). Each chamber was ventilated and temperature controlled with air inlets and outlets. In addition, since each chamber was equipped with a honeycomb waveguide called “cutoff” that prevents the passage and propagation of RF waves, there was no leakage to the outside. Lighting was provided by two LEDs in each chamber, controlled by a self-contained timer (Bachmann GmbH, Germany), providing a 12–12 h light/dark cycle. A management system controlled the exposure schedule, the type and level of exposure signal in each chamber, and measured and recorded exposure parameters, agitator rotation speed, and environmental parameters (airflow, temperature, humidity, and lighting). Mouse dosimetry at 900 MHz RF in these reverberation chambers was performed by the IT’IS Foundation. Dose assessment was performed using the finite-difference time-domain (FDTD) method, as previously described [33].

In this experimental design, two chambers were exposed to RF signals of 900 MHz CW radiation, 0.1 W/kg, and 0.4 W/kg for 3 and 7 days, 1 h, twice a day, in the real world. The other chamber was used as a sham under similar conditions, but without radiation exposure. The experiment was conducted in a blind manner to minimize bias and ensure unbiased data collection and analysis.

### 4.4. Tissue Collection

Inguinal white adipose tissue was dissected from the mice. Each tissue was placed in 2 mL joined neck tubes (Dominique Dutscher) containing RNAlater solution (Thermo Fisher) to prevent RNA degradation. The tissues were then weighed and ground in 1 mL QIazol Lysis Reagent. This grinding was performed in tubes containing RTL Plus Buffer solution (0.6 mL) and 2.8 mm ceramic beads (Qiagen, Germany). The homogenate was incubated at room temperature for 5 min; then, 200 µL of chloroform was added, shaken vigorously for 15 s, and incubated for 2–3 min at room temperature. Total RNA from the samples was extracted using the RNeasy mini kit (Qiagen, Germany) following the manufacturer’s instructions. The amount of total RNA extracted was determined using a Nanodrop spectrophotometer. The eluted RNA was stored at −80 °C until use.

### 4.5. Reverse Transcription

The extracted RNAs were reverse transcribed to complementary DNA (cDNA), using the High-Capacity cDNA Reverse Transcription Kit and adding RNAse inhibitor (Thermo Fisher Scientific, Illkirch, France) according to the manufacturer’s instructions. The reactions were incubated at 60 °C for 60 min using a thermal cycler (Veritipro). The resulting cDNAs were then frozen at −20 °C until use.

### 4.6. Real-Time PCR

Real-time PCR was performed to analyze the expression levels of the target genes. The Taqman Advanced Fast MasterMix kit (Fisher Scientific, Illkirch, France) was used with Taqman Gene expression assays (FAM) for S100b (Mm00485897_m1), Slc6a8 (Mm00506023_m1), DIO2 (Mm00515664_m1), UCP1 (Mm01244861_m1), CPT1a (Mm01231183_m1), Pparα (Mm00440939_m1), ND1 (Mm04225274_s1), Rpph1 (Mm05846955_s1), ACO1 (Mm00801417_m1), NDUFS8 (Mm00523063_m1)), Citrate synthase (Mm00466043_m1), PRDM16 (Mm00712556_m1), 18s RNA (Mm03928990_g1), and GAPDH (Mm99999915_g1) (Fisher Scientific, France). Quantitative PCR was performed in 96-well plates on a thermal cycler. Amplification was performed for 40 cycles. After an initial start of 120 s at 50 °C and a second step of 20 s at 95 °C, each cycle consisted of 1 s denaturation at 95 °C and 20 s annealing at 60 °C. All samples were placed in duplicate wells. Analysis of the PCR results was performed by the 2^−ΔΔCt^ method. Two housekeeping genes (Gapdh and 18s rRNA) were tested under the same conditions. Expression stability was determined using RefFinder (Xie et al., 2012), BestKeeper [34], Normfinder [35], and Genorm [36] programs, with the help of the delta-Ct method (Silver et al., 2006). After a comparison of the results, *Gapdh* was selected as the best housekeeping gene (Appendix A) in white adipose tissue, and 18s rRNA) in brown adipose tissue. The comparative ΔCt method mRNA amounts were estimated using the 2^−ΔΔCT^ method (ΔΔCt = ΔCt_exposed_ − mean ΔCt_control_), relative to Gapdh in white adipose tissue and 18s rRNA in brown adipose tissue.

### 4.7. Measurement of mtDNA-CN

mtDNA-CN was determined using real-time qPCR assay. Briefly, the cycle threshold (Ct) value of a mitochondrial-specific (ND1) and nuclear-specific (RPPH1) target were determined in duplicates for each sample. The difference in Ct values (ΔCt) for each replicate represents a raw relative measure of mtDNA-CN.

### 4.8. Data Processing and Statistical Analysis

Statistical analyses were performed using GraphPad Prism version 7.04 for Windows (GraphPad Software, San Diego, CA, USA). The mRNA expression levels of the target genes were analyzed using the 2^−ΔΔCt^ method. The data are presented as mean ± standard error mean (SEM). mRNA levels have been analyzed by two-tail Mann-Whitney U Tests. For all tests, a *p*-value of less than 0.05 was considered significant.

## Figures and Tables

**Figure 1 ijms-24-10628-f001:**
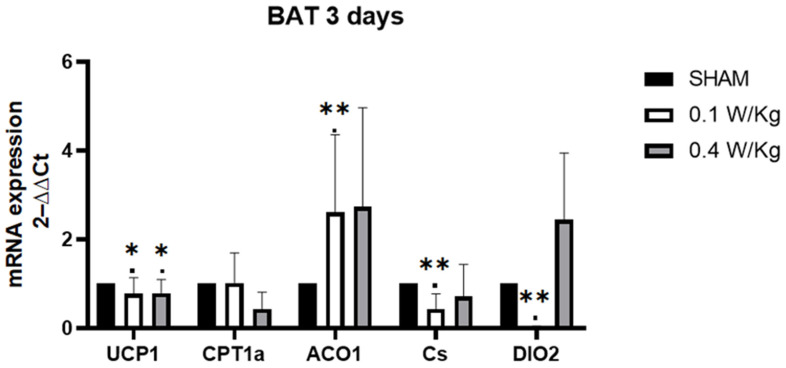
Effect of 3 days of RF exposure on the relative mRNA expression of genes coding for thermoregulation and mitochondrial signaling pathways, measured by qPCR in brown adipose tissue (BAT) of 3-month-old C57BL/6J mice. Values are expressed as fold induction over the SHAM group set at 1. 18S gene is used as house-keeping gene. Data are expressed as mean ± s.e.m (n = 6 mice), * *p* < 0.05, ** *p* < 0.01, two-tailed Mann-Whitney U tests using the 2^−ΔΔCt^ calculation method.

**Figure 2 ijms-24-10628-f002:**
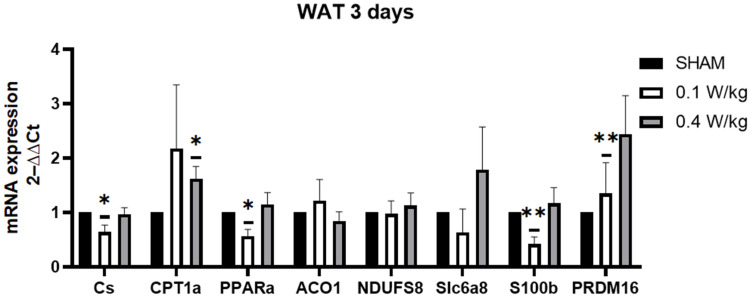
Effect of 3 days of RF exposure on the relative mRNA expression of genes coding for thermoregulation and mitochondrial signaling pathways, measured by qPCR in WAT of 3-month-old C57BL/6J mice. Values are expressed as fold induction over the SHAM group set at 1. GAPDH gene is used as house-keeping gene. Data are expressed as mean ± s.e.m (n = 6 mice), * *p* < 0.05, ** *p* < 0.01, two-tailed Mann-Whitney U tests using the 2^−ΔΔCt^ calculation method2.3. Formatting of Mathematical Components.

**Figure 3 ijms-24-10628-f003:**
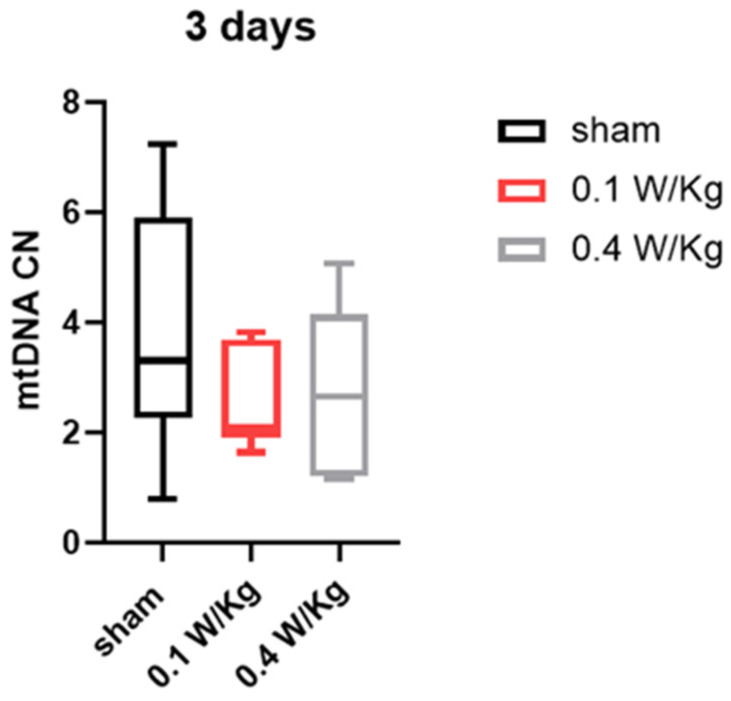
Effect of three days of RF exposure on mitochondrial DNA copy number. Data are expressed as mean ± s.e.m (n = 6 mice), two-tailed Mann-Whitney U tests.

**Figure 4 ijms-24-10628-f004:**
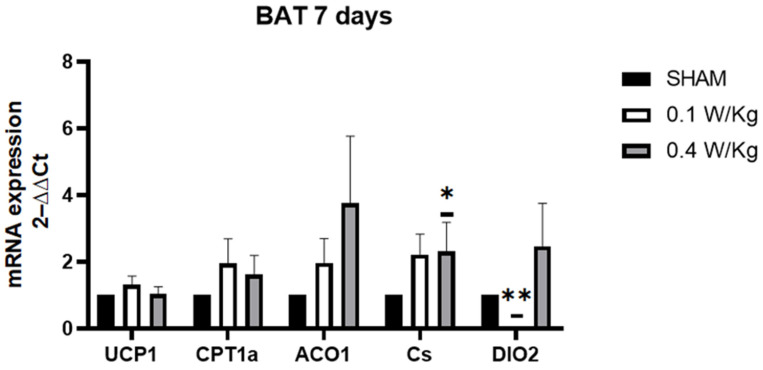
Effect of 7 days of RF exposure on the relative mRNA expression of genes coding for thermoregulation and mitochondrial signaling pathways, measured by qPCR BAT of 3-month-old C57BL/6J mice. Values are expressed as fold induction over the SHAM group set at 1. 18S gene is used as house-keeping gene. Data are expressed as mean ± s.e.m (n = 6 mice), * *p* < 0.05, ** *p* < 0.01, two-tailed Mann-Whitney U tests using the 2^−ΔΔCt^ calculation method.

**Figure 5 ijms-24-10628-f005:**
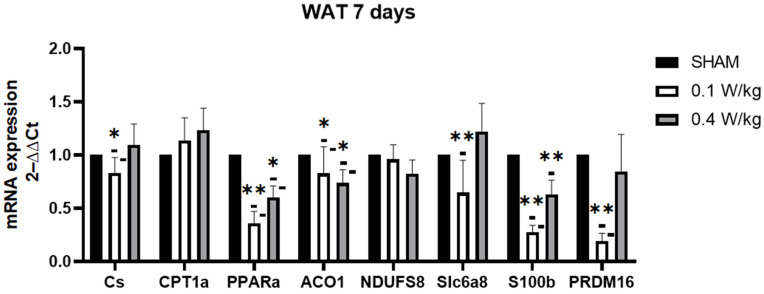
Effect of 7 days of RF exposure on the relative mRNA expression of genes coding for thermoregulation and mitochondrial signaling pathways, measured by qPCR in WAT of 3-month-old C57BL/6J mice. Values are expressed as fold induction over the SHAM group set at 1. GAPDH gene is used as house-keeping gene. Data are expressed as mean ± s.e.m (n = 6 mice), * *p* < 0.05, ** *p* < 0.01, two-tailed Mann-Whitney U tests using the 2^−ΔΔCt^ calculation method.

**Figure 6 ijms-24-10628-f006:**
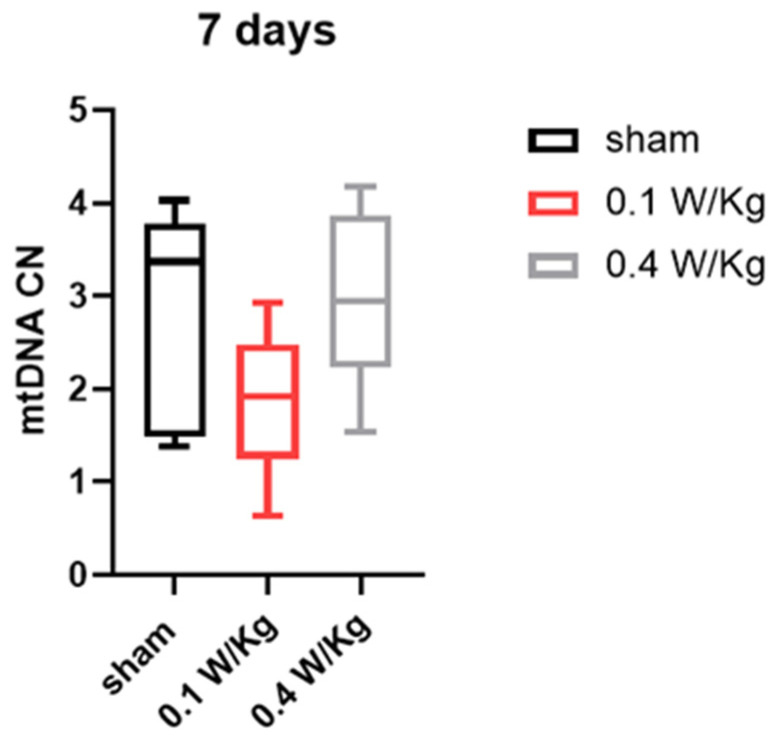
Effect of seven days of RF exposure on mitochondrial DNA copy number. Data are expressed as mean ± s.e.m (n = 6 mice), two-tailed Mann-Whitney U tests.

## Data Availability

The data used in this study are available upon request. Please contact the corresponding author for access to the data.

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
