# Peer review of "Dose- and Time-Dependent Effects of Radiofrequency Electromagnetic Field on Adipose Tissue: Implications of Thermoregulation and Mitochondrial Signaling"

_ijms, 2023, doi:10.3390/ijms241310628_

Round 1
Reviewer 1 Report
Thank you for your great manuscript about "Dose-Response Effects of Radiofrequency Radiation on Adipose Tissue: Implications of Thermoregulation and Mitochondrial Signaling"
It's very interesting and useful.
please insert some details of the differencies between BAT and WAT and why it is occured you think.
Reviewer 2 Report
The manuscript Jennifer Maalouf investigated the effects of low-intensity radiofrequency (RF) fields on the expression of adipose genes involved in thermoregulatory and mitochondrial functions. Using the reverberation chamber system, RF signal generator and amplifier, male mice were exposed to continuous RF field at 900 MHz, with specific absorption rates (SAR) of 0.1 or 0.4 W/kg, 1 hour duration, twice a day, for 3 or 7 consecutive days. The author found the downregulation of uncoupling protein 1 (UCP1), citrate synthase (CS), and type 2 iodothyronine deiodinase (DIO2), and upregulation of cytosolic aconitase 1 (ACO1), and suggested that the possible reduction of thermogenesis and mitochondrial activity might occur in BAT after 3 days of RF exposure. Those effects were partially compensated after 7 days of RF exposure. Similarly, decreased expression of PPARα, PRDM16, CS and a calcium binding protein S100B were observed in WAT, suggesting the possible disruptions of fatty acid oxidation, browning, and thermogenesis after 3 to 7 days of RF exposure. Overall, this manuscript presents the important data that contribute to further understanding of the complex effects of RF exposure on adipose tissue energy metabolism. Thus, I would like to recommend the manuscript for publication after minor revision.
Specific comments
· The author should explain why a specific RF wavelength (900 MHz), SAR values (0.1 and 0.4 W/kg) and exposure duration (1 hour) were chosen to study the dose-response effects of RF on adipose tissues.
· The author only investigated the gene expression to study the changes of adipose tissue metabolism and thermoregulation. It will be better if the author also monitors the actual changes of mitochondrial metabolism, browning and thermogenesis (e.g., body temperature).
· Abbreviations must be spelled out on the first appearance in text. E.g., NDUFS8.
· Typos must be checked and corrected. E.g., “PRDM16” in Figure 5, and line 182.
Nil
Reviewer 3 Report
Review on Dose-Response Effects of Radiofrequency Radiation on Adipose Tissue: Implications of Thermoregulation and Mitochondrial Signaling
Comments for authors for major revisions
Comment 1: Revise the title. The term 'dose-response effects' seems inadequate; instead, it would be more appropriate to refer to 'dose and time-dependent effects.' Similarly, the phrase 'radiofrequency radiations' does not accurately capture the concept.
Comment 2: Revise the statements in the abstract section “were exposed or not to RF fields at 900 MHz” in lines 15 – 16, and “three or seven days” in line 17. These statements need to correct. The words “exposed or not” and “three or seven days” seems confusing.
Comment 2: It is important to describe how electromagnetic radiation affects biological systems. I encourage authors to add some background on this topic. The suggested article may assist authors in expanding their background knowledge and understanding the mechanisms by which the electromagnetic field interacts with and affects biological systems for various effects. The inclusion of this recent article could help to strengthen the introduction section.
Article: Microwave Radiation and the Brain: Mechanisms, Current Status, and Future Prospects. International Journal of Molecular Sciences vol. 23 (2022). [https://doi.org/10.3390/ijms23169288].
Comment 3: The introduction section should be concise and focused. I recommend shortening the descriptive text, as it is not essential for this manuscript. Instead, it is advisable to concentrate on the topic's motivation and provide relevant background information.
Comment 4: Understanding the properties of the radiations is crucial as their effects strongly depend on them. Therefore, it is essential to provide a detailed explanation of the exposure system and the specific properties of the radiation.
Comment 5: Given that the electromagnetic radiation exposure occurs continuously for one hour, twice a day, there is a concern regarding any potential temperature rise. It is important to ascertain whether the authors conducted measurements to determine the temperature after the exposure.
Comment 6: Describe how the authors calculated the electromagnetic energy in (W/kg), it is important to note that power is typically measured in watts (W) in RF systems. The manuscript should provide more details regarding the specific methodology used by the authors to derive the energy values in W/kg. Additionally, it is necessary to explain how the authors adjusted the electromagnetic energy to reach the targeted levels of 0.1 W/kg and 0.4 W/kg within the exposure system.
Comment 7: The manuscript inadequately describes the role of reactive oxygen species (ROS) in response to the exposure of RF fields. Given that ROS plays a critical role in inducing biological effects, it is crucial to enhance the discussion surrounding ROS. To improve the manuscript, it is recommended to incorporate the findings and insights from the following articles, which provide valuable information on the topic:
a. https://doi.org/10.1016/j.jmau.2017.07.003
b. https://doi.org/10.1080/15368378.2019.1656645
c. https://doi.org/10.3389/fcell.2023.1067861
d. https://doi.org/10.1021/acs.nanolett.9b01735
Not limited to these only, authors should find related articles to include in their discussion part.
Comment 8: Include the conclusion section.
Comment 9: The paper contains errors and typos that make it difficult to understand and distort its intended meaning. I encourage authors to reread carefully and fix any grammatical errors.
The paper contains errors and typos that make it difficult to understand and distort its intended meaning. I encourage authors to reread carefully and fix any grammatical errors.
Round 2
Reviewer 3 Report
I appreciate that the authors have addressed my comments and concerns in the revised draft. I recommend publishing this paper in its present form.